# Incremental Dynamic Analysis for Estimating Seismic Performance of Multi-Story Buildings with Butterfly-Shaped Structural Dampers

**Alireza Farzampour [1], Iman Mansouri [2,\*] and Hamzeh Dehghani [3]**

[1] Department of Civil and Environmental Engineering, Virginia Tech, Blacksburg, VA 24061, USA; afarzam@vt.edu

[2] Department of Civil Engineering, Birjand University of Technology, Birjand 97175-569, Iran

[3] Department of Civil Engineering, Higher Education Complex of Bam, Bam 14477-76613, Iran; hdehghani@bam.ac.ir

\* Correspondence: mansouri@birjandut.ac.ir

**Abstract:** Structural strength and stiffness were previously investigated to sufficiently improve the lateral load resistance against major events. Many buildings require appropriate design to effectively withstand the lateral seismic loads and reduce the corresponding damages. Design methodologies and structural elements were recently introduced to improve the energy dissipation capability and limit the high force demands under seismic loadings. The new systems are designed to protect the structural integrity and concentrate the inelasticity in a specific area, while the remaining parts are kept undamaged and intact. This study introduces a new structural system with dampers having strategic cutouts, leaving butterfly-shaped shear dampers for dominating the yielding mechanism over other brittle limit states. The new system is designed for re-establishing the conventional eccentrically braced frame system with simple linking beams. The system with strategic cutouts is subsequently used and compared with the eccentrically braced frames (EBF) system for seismic performance investigation and incremental dynamic analysis (IDA), using the OpenSees program, which is used to indicate the collapse behavior under forty-four selected ground motions. Results show that the butterfly-shaped multi-story buildings, compared to the corresponding conventional systems, are capable of enhancing the system resistance against lateral seismic loads by postponing the collapse state to the larger drift ratio values.

**Keywords:** incremental dynamic analysis; eccentrically braced frame; butterfly-shaped shear damper; hysteretic behavior; shear link

## 1. Introduction

Structural fuses were recently proposed to concentrate damages and inelasticity at a specific part of a building, which saves the critical members from significant damages [1,2]. These fuses, in various shapes and sizes, allow inelastic displacement, by a yielding mechanism, in high drift ratios [3,4], which protect the surrounding members and improve the structural integrity under high seismic forces. A common type of structural fuse is made by the strategic removal of the material, leaving a structural fuse system in which the ductile mechanisms occur over the brittle limit states under shear loading [5–7]. A typical structural element is a butterfly-shaped fuse, with linearly varying width, along the length of the link to efficiently equalize the demand and capacity moments. Figure 1 indicates a typical butterfly-shaped fuse system used in structural applications.

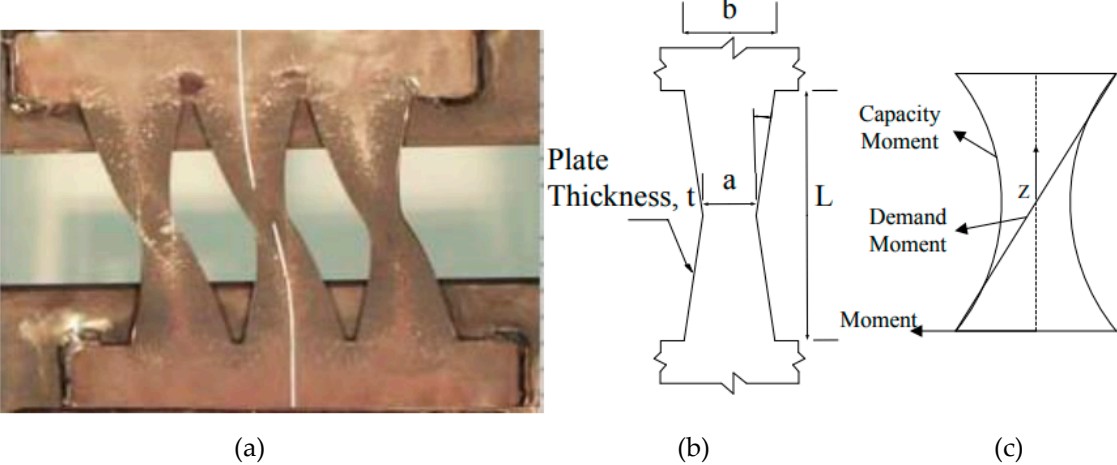

(a)                                  (b)                                  (c)

**Figure 1.** The butterfly-shaped hysteretic damper. (**a**) Butterfly-shaped plate (from [8]); (**b**) Geometry; (**c**) Moment diagram.

The butterfly-shaped fuse system was previously used for fortification and structural performance improvement in multi-story buildings (Figure 2). These fuses are added to the structure for controlled yielding over the brittle limit state and seismic resistance features. In addition, implementation of the butterfly-shaped fuses could effectively improve the initial stiffness, the strength of the system, and prevent secondary damages [9,10]. Having yielding hinges far from the discontinuities, by appropriate geometrical design methodologies in butterfly-shaped dampers, results in full hysteretic behavior, controlled limit states, and uniform stress distribution along the length of the damper [11–13].

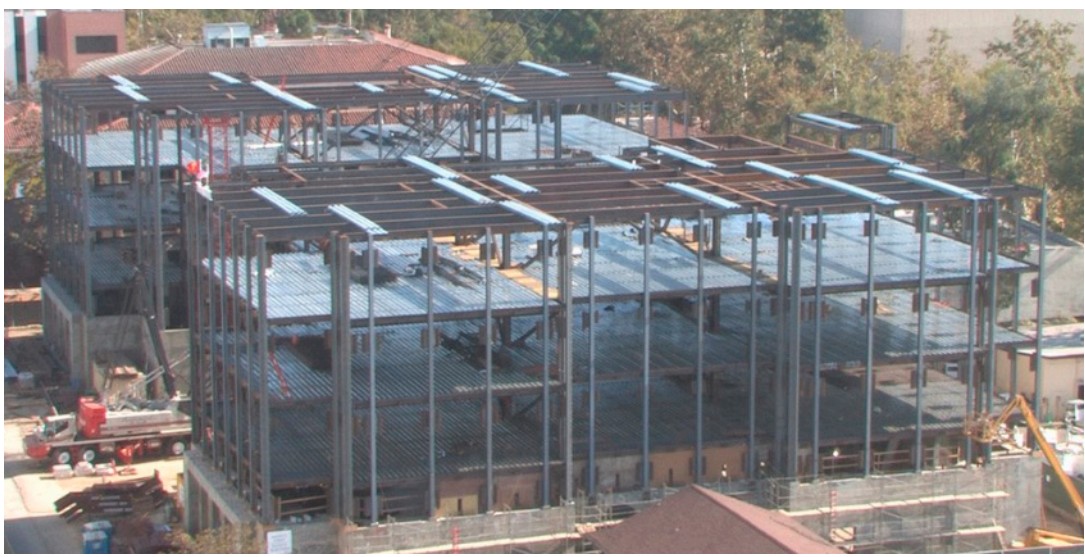

(a)

**Figure 2.** *Cont.*

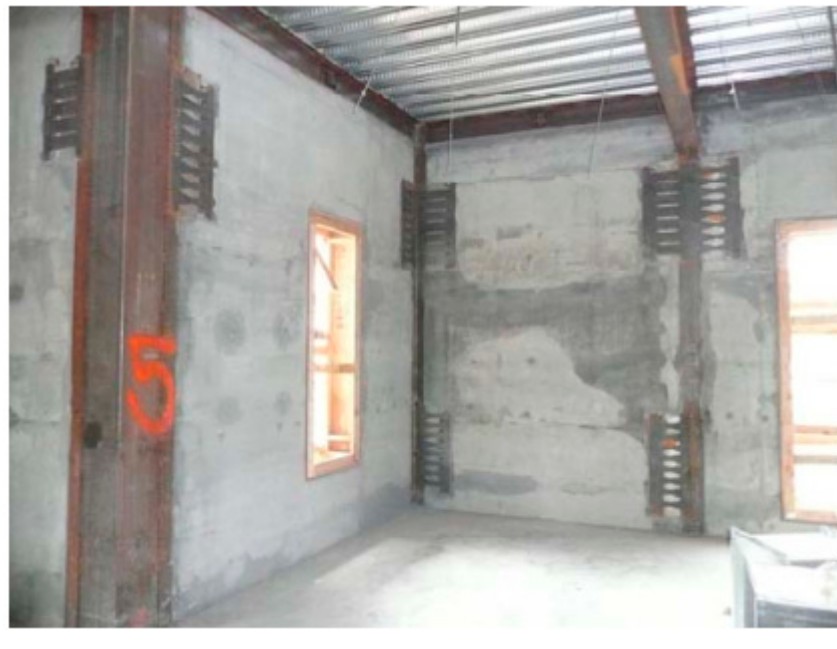

(b)

**Figure 2.** Implementation of the butterfly-shaped hysteretic damper. (**a**) Butterfly-shaped fuses in multi-story buildings; (**b**) Butterfly-shaped fuses used to enhance energy dissipation (from [8]).

Many studies show that the strategic removal of the materiel could prevent and postpone the brittle limit states, especially buckling, and promote the yielding mechanism [3]. The resulting replaceable fuse system is generally designed and implemented in highly seismic affected areas for increasing the whole system's energy dissipation [14,15]. The energy dissipation comparison of the structures with shear links could be more than 5 times larger than the conventional systems for a specific story under the seismic load condition [3]. Additionally, to reduce the demand force on the connection areas [16], uniformly distribute the imposed stresses [17], and localize the yielding in the desirable parts of the structure, seismic structural fuses, in various forms and shapes, are introduced [6,8]. The detailed comparisons of the structural fuse systems are necessary for understanding the advantages of new structural fuse systems for use in multi-story buildings.

In this study, the effects of the seismic structural fuses system with butterfly-shaped dampers for multi-story structures are investigated and compared with conventional systems. Two systems are designed based on the Structural Engineers Association of California (SEAOC) manuals [18,19] for seismic investigation purposes. Incremental dynamic analysis is incorporated as one of the most efficient computational seismic analysis methods to perform the seismic assessment for understanding the behavior of structures. The seismic investigations are subsequently conducted by an incremental dynamic analysis study to indicate the collapse behavior under forty-four selected ground motions, based on accelerograms generating methodologies proposed by FEMA P751 [20], and the advantages of the new structural fuse system over the conventional structures are shown in detail. The mean response spectrum of all accelerograms used for analysis purposes is shown in Figure 3. The record set is related to far-field, which does not include the vertical component of ground motion, since this direction of earthquake shaking is not considered of primary importance for collapse evaluation and is not required by the proposed methodology [20] for nonlinear dynamic analysis.

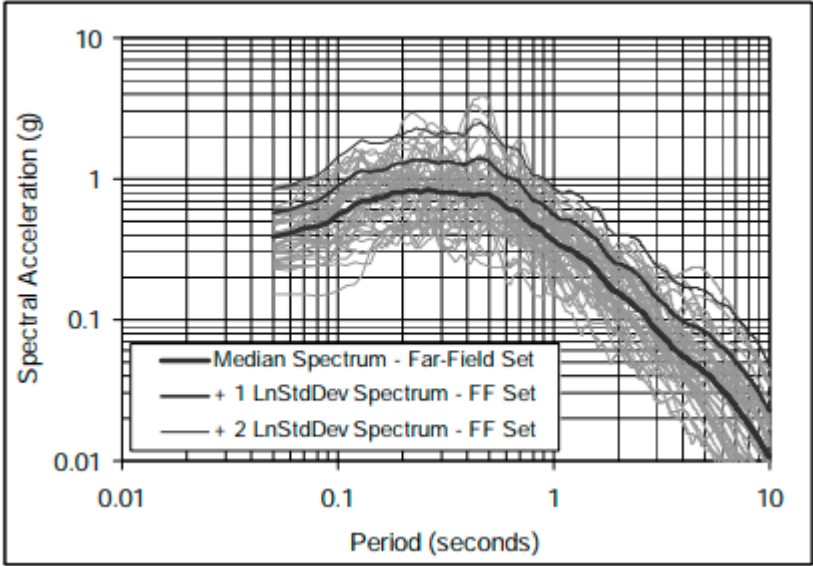

**Figure 3.** Spectrum of all accelerograms used for incremental dynamic study [20].

## 2. Design and Verification of the Modeling Methodology

Eccentrically braced frames (EBFs) can be detailed to provide ductile behavior under a severe earthquake [21–26]. To investigate the seismic behavior of the multi-story system with conventional fuses and butterfly-shaped dampers, the prototype example from SEAOC [18] is considered, based on which, the OpenSees software is used, subsequently, for incremental dynamic analysis. This example is related to a six-story structure with an EBF system as the main lateral resistance system, which is used for the design of a six-story butterfly-shaped fuse structure. Figure 4 schematically represents the two investigated structures. The conventional structures, with eccentrically braced frames and a butterfly-shaped fuse system, are shown in Figure 4a,b, respectively. Figure 4c shows the butterfly-shaped fuse system designed separately at each story level and Figure 4d shows butterfly-shaped dampers used in multi-story buildings.

The design forces are indicated in detail in Table 1. The frame beams and columns in both conventionally and butterfly-shaped fuse systems are considered to be the same and, specifically, the linking beam at each story level is designed separately, based on shear link design guideline procedures [3]. It is noted that the shear at butterfly-shaped fuses is calculated based on the geometrical properties of the system, which is summarized in Table 1 and extracted from a SEAOC [18] EBF example. Figure 5 shows the design groups categorized according to Table 1. The butterfly-shaped fuse geometry is designed based on the analyzed shear for each story level.

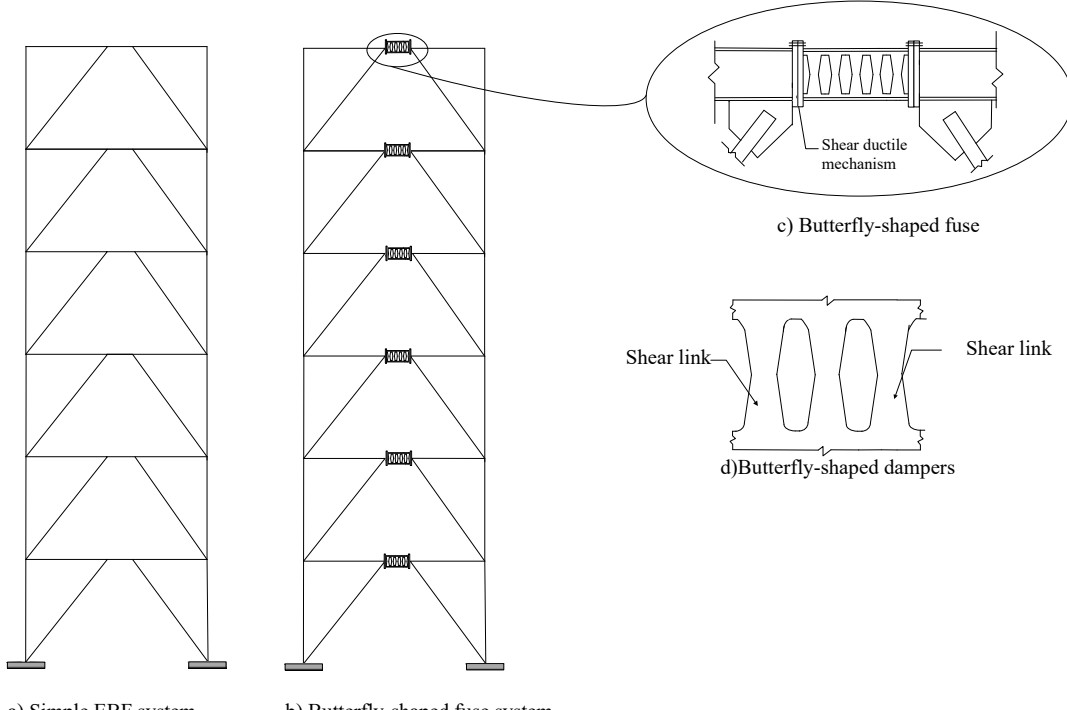

a) Simple EBF system    b) Butterfly-shaped fuse system

**Figure 4.** Schematic representation of a six-story building with and without butterfly-shaped fuses for IDA analysis. (**a**) Simple EBF system; (**b**) Butterfly-shaped fuse system; (**c**) Butterfly-shaped fuse; (**d**) Butterfly-shaped dampers.

**Table 1.** Shear forces used for designing butterfly-shaped shear fuses.

| Level | Shear (kips) | Cumulative Force (kips) | Design Section for EBF | Design Force for the BF Links | | Design Groups for BF System |
|---|---|---|---|---|---|---|
| Roof | 60 | 60 | BU 13 × 53 | 244 | 244 | III |
| 6th | 60 | 120 | BU 13 × 53 | 244 | | |
| 5th | 88 | 208 | BU13 × 53 | 358 | 358 | II |
| 4th | 125 | 333 | W 10 × 68 | 508 | 508 | I |
| 3rd | 125 | 458 | W 10 × 68 | 508 | | |
| 2nd | 125 | 583 | W 10 × 68 | 508 | | |

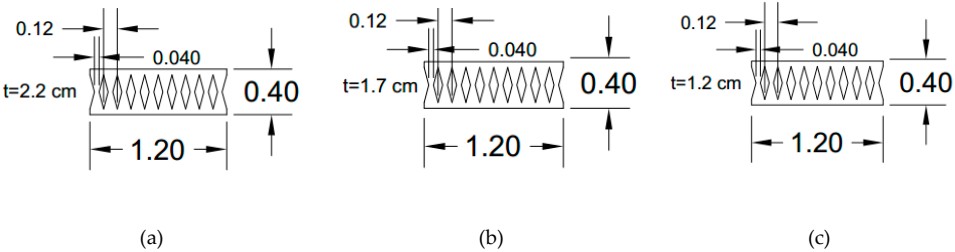

    (a)        (b)        (c)

**Figure 5.** Three design groups used for incremental dynamic analysis (IDA) (thickness is in centimeters and the rest is in meters). (**a**) Category I, for lower story structural fuses based on W 10 × 68 beam dimensions; (**b**) Category II, for middle story structural fuses based on BU13 × 53 beam dimensions; (**c**) Category III, for upper story structural fuses based on BU 13 × 53 beam dimensions.

To verify the computational modeling methodology, two verification studies are conducted. The first verification study is related to an experimental test with an hourglass damper system, computationally modeled in the finite element ABAQUS package. In the second verification study, the OpenSees modeling methodology is verified under the EBF loading protocol condition.

The laboratory test for the first verification study conducted on shear fuses is verified with finite element analysis methodology. The experimental study was conducted by Aschheim and Halterman [17] on a beam with circular cutouts. To verify the modeling methodology, a C3D20R element is used for having high accuracy and for avoiding hourglassing and shear locking effects. A bi-linear stress-strain material is assumed with the yield strength of 379 MPa, an elastic modulus of 200 GPa, and a strain-hardening modulus of 1.38 GPa. According to the geometry of the frame and the beam, the story shear is captured by multiplying 1.43 times the beam shear retrieved from finite element analysis (FEA) and story drift is calculated based on the beam chord rotation divided by 1.34. The shear strength of the beam, before and after buckling limit state occurrence, is reported to be 76 (kN) and 57.4 (kN), while the computational models were able to capture the buckling at the specified drift ratio with 73.2 (kN) and 52.7 (kN), respectively. This indicates that the computational FEA models are able to accurately capture the behavior of the experimental laboratory test and the hysteretic pushover behavior, which is shown in Figure 6.

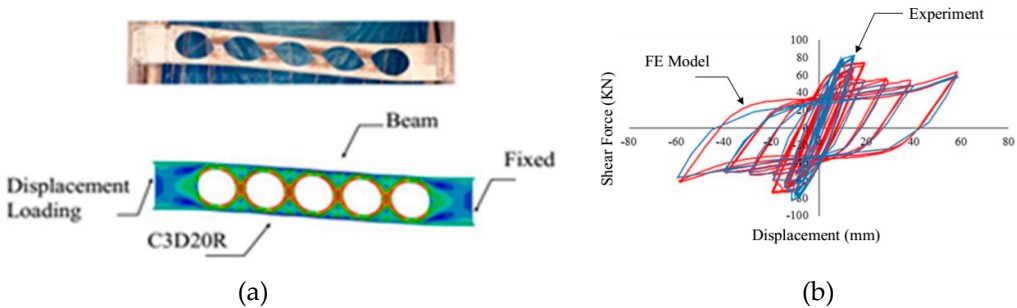

(a)  (b)

**Figure 6.** Verification of the finite element modeling methodology against laboratory test [17]. (**a**) FEA model; (**b**) Comparison of FEA with test.

The second verification is conducted for calibrating fuse behavior of the used reduced models in OpenSees, based on which the IDA is conducted. This verification indicates that computational models in OpenSees are able to predict the results of the multistory EBF structures with sufficient accuracy.

This verification study is conducted to investigate the OpenSees computation models and compare them with FEA results. For this purpose, two similar systems, schematically shown in Figure 7a, are modeled separately in each software and are subjected to the AISC 341 [27] loading protocol for EBF systems. The computational modeling methodology is considered based on reference [3], in which the butterfly-shaped dampers are modeled with a displacement-based beam element. The hysteretic results are compared and it is concluded that OpenSees models used for multi-story structure modeling have more than 98% accuracy in capturing the behavior of the butterfly-shaped shear dampers, which is shown in Figure 7.

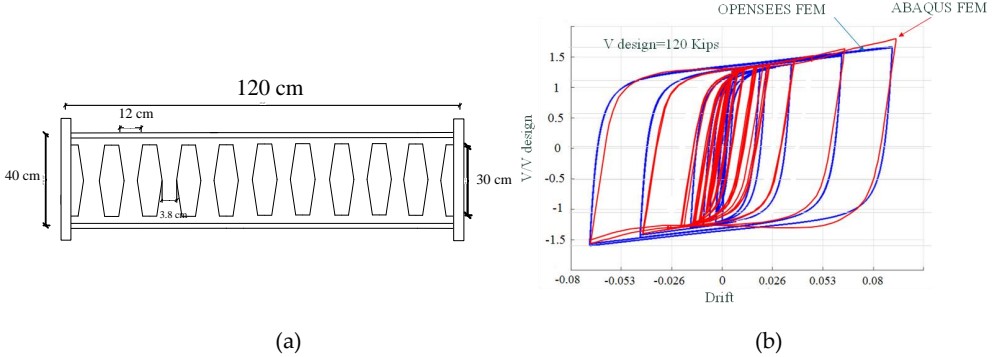

(a)  (b)

**Figure 7.** Verification of OpenSees modeling methodology with FEA results based on AISC 341 loading protocol for EBF. (**a**) Butterfly-shaped beam; (**b**) Verification study.

## 3. Comparison of the Seismic Behavior of the EBF Conventional and Butterfly-Shaped Systems

To initially indicate the behavior of the conventional and butterfly-shaped fuse buildings, static pushover analysis is conducted on both systems. The displacement controlled loading is proportionally applied to each story level. The total shear at the bottom and last story displacement are monitored. Figure 8 shows that the two systems have similar pushover curves and, for both systems, stiffness degradation happens at the same displacement value. It is worthy of notice that both systems are designed with a 2% damping ratio, which is applied to the models with Rayleigh damping methodology. It is concluded that both systems have similar initial stiffness and ultimate strength. The allowable drift ratio recommended by ASCE 07 [28] is checked subsequently to be less than a 0.02 drift ratio for each story level.

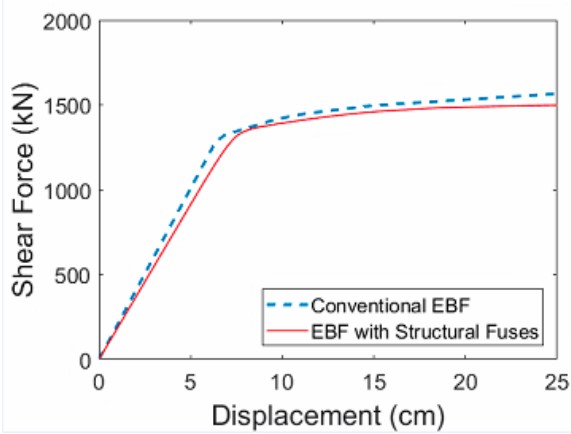

**Figure 8.** Pushover curves comparison of the two systems.

In addition, the Eigenvalues analysis for the both systems is summarized in Table 2, which shows that for both of the systems, the first and second period are close. This shows that stiffness of the two systems are close since the mass is the same, based on the similar seismic weight associated with each story level.

**Table 2.** Comparison of the periods of EBF building with BF fuse building (sec).

| Mode No. | BF Fuse System | Conventional System |
|:---:|:---:|:---:|
| 1 | 1.332 | 1.283 |
| 2 | 0.518 | 0.482 |
| 3 | 0.299 | 0.289 |
| 4 | 0.243 | 0.227 |
| 5 | 0.192 | 0.18 |

For seismic investigations, in this study, incremental dynamic analysis (IDA) for earthquake engineering assessment is conducted to thoroughly investigate the seismic behavior of the EBF conventional and BF systems. The results of this study could be implemented in the risk analysis of the new fuse systems. IDA includes multiple nonlinear dynamic analyses of a specific prototype structure under a group of ground motion records, which are typically scaled to several seismic intensity levels. For this propose, the scale factors from 0.1 g up to 4.0 g are applied to both systems, with an accelerations interval of 0.1 g. The scaled ground motions are applied to the structure to capture seismic behavior from elastic regions to inelastic regions and, ultimately, collapse occurrence. The results are subsequently represented by intensity measure versus structural response.

In this study, the structural response is the maximum fuse shear occurred under the applied ground motions. For each applied scaled ground motion, all the stories' shear angles are monitored and the maximum shear angle, considering all the stories, is captured. Then, the peak ground acceleration,

as the selected intensity measure, is derived and drawn with the corresponding maximum shear angle. Forty-four ground motions [20] are selected and for each ground motion a total of 40 scale factors are applied.

The results for the two systems are summarized in Figures 9 and 10 for 3520 runs related to the conventional and butterfly-shaped fuse structures. From Figures 9 and 10, it is concluded that, in the majority of the cases, more than 50% of the stiffness is reduced by the scale factor of 2.85 g for the conventional EBF system, while for the butterfly-shaped fuse system, the corresponding scale factor was 3.05 g. This indicated that, on average, the system with butterfly-shaped fuses is able to resist against higher ground motion intensity, compared to corresponding conventional EBF system. In addition, considering Figure 9 with Figure 10, it is shown that, for the majority the scaled ground motions, the butterfly-shaped system has lower shear angle values, compared to the conventional EBF system. Having lower shear angles causes less probability of fracture and collapse under seismic loading with high magnitudes, since higher shear angles lead to larger drifts, which increase the possibility of fracture.

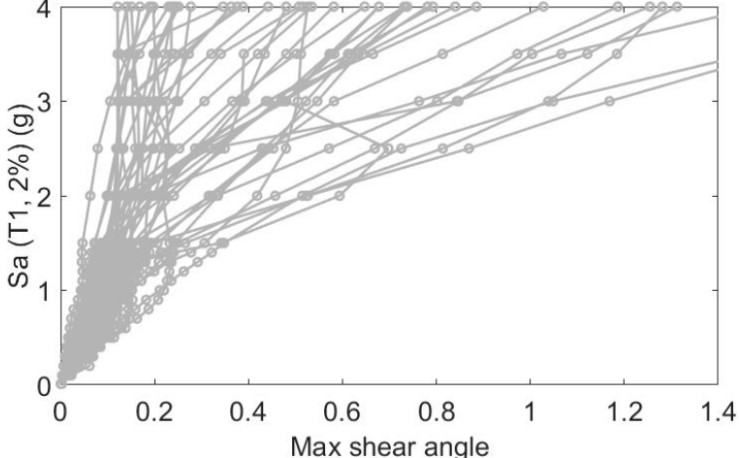

**Figure 9.** Incremental dynamic analysis for an EBF system.

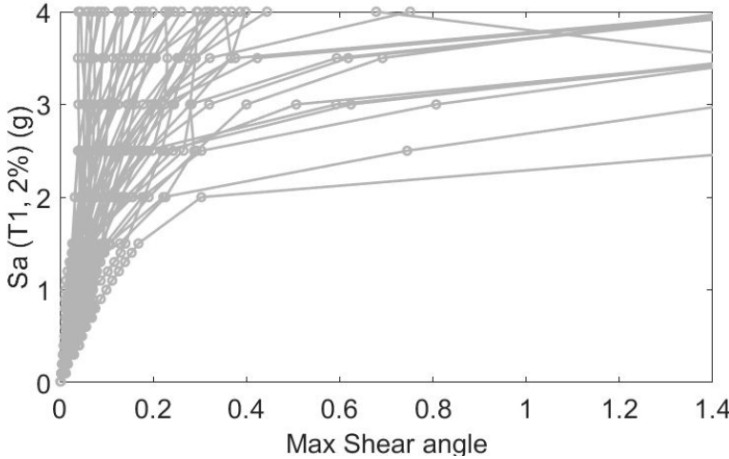

**Figure 10.** Incremental dynamic analysis for a butterfly-shaped fuse system.

## 4. Conclusions

The strategic removal of the material to shift from the brittle limit state occurrence to a ductile yielding mechanism would improve the hysteretic behavior and reduce the fracture possibility. In this research, two prototype buildings are investigated to compare the seismic performance of a conventional EBF system with a new system with butterfly-shaped shear dampers. The models are built

in computational software, after verifying the modeling methodology and the reduced order models. The effect of having butterfly-shaped dampers for multi-story structures is seismically investigated. The incremental dynamic analysis is used as a powerful tool for understanding the seismic behavior of the multi-story structures. It is shown that multi-story buildings with butterfly-shaped dampers are able to effectively increase the critical ground motion scale level, in which 50% of the initial stiffness is eliminated, and enhance the system resistance against a lateral seismic load by postponing the collapse state to a larger drift ratio value. It is shown that, in general, the shear angles associated with forty-four prescribed ground motions, with different scales, would be significantly lower than the corresponding conventional system, which yields to more ductile performance and full hysteretic response at each story level.

**Author Contributions:** Conceptualization, A.F., I.M., H.D.; Methodology, A.F.; Software, A.F.; Validation A.F.; Formal Analysis A.F.; Investigation, A.F.; Resources, A.F.; Data Curation, A.F; Writing-Original Draft Preparation, A.F. and I.M.; Writing-Review & Editing, A.F. and I.M.; Visualization, A.F.; Supervision, A.F.; Project Administration, A.F.; Funding Acquisition, A.F.

**Funding:** This research received no external funding.

**Conflicts of Interest:** The authors declare no conflict of interest.

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
