# Peer review of "Incremental Dynamic Analysis for Estimating Seismic Performance of Multi-Story Buildings with Butterfly-Shaped Structural Dampers"

_buildings, doi:10.3390/buildings9040078_

Round 1
Reviewer 1 Report
The paper addresses an interesting issue, which is the use of butterfly-shaped structural dampers to improve the seismic behaviour of buildings, reducing damage in critical structural members.
There are 28 references in the text, being about 61% from the last 5 years, and only about 14% are more than 10 years old, which is ok.
The subject of the paper is interesting for publishing; however, I believe that the present version of the paper is not suitable for publishing, and a major revision is needed.
Overall, the paper is very confusing, being difficult to understand the logic of the study.
In line 25, where is "EBF", it should be "eccentrically braced frames (EBF)", otherwise anyone that only reads the abstract might not understand the meaning of EBF.
A careful revision of the English grammar should be carried out. For example, in line 65 where is “understating” probably the authors meant to write “understanding”. In line 179, where is written “butterfly0shaped” it should be “butterfly-shaped” instead.
The authors must present the mean response spectrum of all accelerograms that where used in the analysis, and to clarify how these accelerograms were selected, because it is said in line 73 that the selected ground motions were “recommended by FEMA P751”. I couldn’t find any recommendation for using a specific set accelerograms, and only methodologies for generating those accelerograms are presented. The reader might understand that the FEMA P751 proposes a given set of 44 ground motions, which I believe it was not the authors objective.
It becomes confusing about which software was used for all the analysis, because that it is not well explained. It seems that the authors used the OpenSees software, and that was the reason to carry out a comparison between OpenSees and ABAQUS results. However, the way this matter is presented in the paper is seems confusing to me, and it should be rewritten.
The use of OpenSees should be clarified, beginning in the introduction.
I also believe that Figures 8 and 9 don’t give to the reader a good perception of the effectiveness of the butterfly-shaped. A better discussion should be presented.
What it is really important is to compare the total amount of dissipated energy on each structural system (namely by comparing ratios between the results obtained with the two studied systems), and to find out what is the amount of damage that is reduced in critical structural members using the butterfly-shaped, which is the first goal that is presented in lines 34 and 35 of the introduction.
Author Response
Please find attached a revised version of our manuscript “Incremental dynamic analysis for estimating the seismic performance of multi-story buildings with butterfly-shaped structural dampers”, which we would like to resubmit after major revision for publication as a Regular paper in Buildings.
Your comments and those of the reviewers were highly insightful and enabled us to greatly improve the quality of our manuscript. The original manuscript is also improved grammatically considering the reviewers' comments. In the following pages are our point-by-point responses to each of the comments of the reviewers as well as your own comments.
Revisions in the text are shown using yellow highlight for additions. In accordance with reviewer 1’s, and reviewer 2’s, suggestions, we have revised the following items.
We hope that the revisions in the manuscript and our accompanying responses will be sufficient to make our manuscript suitable for publication in Buildings.
We shall look forward to hearing from you at your earliest convenience.
Yours sincerely,

Reviewer 2 Report
Incremental dynamic analysis for estimating seismic performance of multi-story buildings with butterfly-3 shaped structural dampers
1. The paper seems to be well written and the results are significant as they evaluate a new structural fuses system with butterfly-shaped dampers that could reduce physical vulnerability of buildings to earthquakes.
2. Please make an overall check of the text. Overall the English in good, but there are some small errors: ex. line 193 Two prototype building is are investigated in this research to compare...
Author Response

(The authors gave the same response as above.)

Round 2
Reviewer 1 Report
The paper has been improved. It is now possible to see the dispersion related to the ground motions used in this study.
The description of the adopted methods is now more clarified, and the discussion was also improved.
The authors have addressed most of my concerns.